# Convolutional Learnable-Group Weightless Neural Network

Qinhong Ma[1]   Yulin Chen[2]   Zhiwei Fan[2]   Suzhen Wu[1]   Bo Mao[1]

## Abstract

Weightless Neural Networks (WNNs) based on interconnected Lookup Tables (LUTs) have attracted attention for inference in extremely compact models, but achieving competitive accuracy under such tight resource budgets remains challenging. To address these issues, we introduce the Convolutional Learnable-Group Weightless Neural Network (CLGN). CLGN constructs convolutional layers using LUTs and incorporates a learnable GroupSum connection, thereby enhancing the accuracy of WNNs while maintaining low implementation resource consumption. Moreover, we propose a hierarchical training strategy to improve the training efficiency. We evaluate CLGN in two edge computing scenarios: (1) FPGA, where we evaluate accuracy, latency, throughput, power consumption, LUTs usage, and parameter size; and (2) Microprocessor, where we evaluate latency and memory usage. Compared with the state-of-the-art solutions, the proposed CLGN achieves superior accuracy while maintaining lower implementation resource consumption.

## 1. Introduction

Deep Neural Networks (DNNs) have become the core focus of current AI research, with numerous studies exploring various aspects, including achieving high inference accuracy while pursuing compact model size and resource efficient inference (Desislavov et al., 2021). Many prior works have sought to address this objective through techniques such as model pruning (Dong et al., 2017), quantization (Banner et al., 2018; Chmiel et al., 2021; Faghri et al., 2020; Zhou et al., 2025), Sparse Neural Networks (Sung et al., 2021; Sun et al., 2021; Ma & Niu, 2018), ensemble methods (Weng et al., 2025; Andronic & Constantinides, 2025), and

---

[1]School of Informatics, Xiamen University, Xiamen, China [2]Tianjin University, Tianjin, China. Correspondence to: Bo Mao <maobo@xmu.edu.cn>.

*Proceedings of the 43rd International Conference on Machine Learning*, Seoul, South Korea. PMLR 306, 2026. Copyright 2026 by the author(s).

substitution approaches (Khataei & Bazargan, 2025). However, these methods still fail to fundamentally reduce the model footprint and inference resource requirements, as the underlying DNN architecture remains dominated by large numbers of parameterized multiply accumulate operations.

To overcome this architectural limitation, many multiplication free models (Yayla & Chen, 2022; Chen et al., 2020; Elhoushi et al., 2021; Nguyen et al., 2024; Samragh et al., 2021; Qin et al., 2024; 2022; He et al., 2025; Wang et al., 2019) have been proposed. Among them, one particularly effective architecture is the Weightless Neural Network (WNN) (Susskind et al., 2023). Unlike conventional DNNs that rely on weighted connections, each neuron in a WNN is implemented using a Lookup Table (LUT) with an input width of $n$. This means that each neuron connects to only $n$ inputs (a LUT with $n$ inputs is denoted as LUT-$n$) and produces a single output. The LUT-based design significantly enhances the nonlinearity of WNNs (Aleksander et al., 2009). Furthermore, since WNNs are implemented via LUTs, they are highly compatible with FPGA acceleration. Leveraging the basic logic units of FPGAs to realize trained models can effectively reduce implementation consumption.

However, because WNNs are inherently LUT-based, their discrete and stochastic characteristics prevent the use of conventional backpropagation training methods, making training highly challenging. Traditional WNN training adopts a forward learning scheme, where the model directly memorizes the input patterns. This approach, however, suffers from poor generalization and low inference accuracy. To improve this, the single layer WNN architecture ULEEN (Susskind et al., 2023) integrates gradient descent training and straight through estimators (Bengio et al., 2013), and demonstrates the effectiveness of WNNs on FPGA platforms. ULEEN significantly outperforms Binary Neural Networks (BNNs) (Umuroglu et al., 2017) in terms of power consumption, latency, and circuit area. Nonetheless, ULEEN requires a very wide single layer WNN to achieve higher inference accuracy, which in turn leads to large popcount operations and higher implementation consumption.

Meanwhile, DiffLogicNet (Petersen et al., 2022) and Conv-DiffLogicNet (Petersen et al., 2024) introduced a training strategy based on LUT-2 implementations of WNNs. Each LUT-2 is associated with all possible Boolean functions,

each assigned a corresponding probability. Training optimizes only these probabilities through differentiable relaxation, whereas inference typically selects the Boolean function with the highest probability. However, this approach restricts $n$ to 2, since the number of candidate Boolean functions grows double exponentially with $n$, making the probabilistic training of LUT-$n$ infeasible for larger $n$. This restriction introduces two issues: (1) The VC dimension of WNNs increases exponentially with $n$ (see Appendix E), meaning that a model using many LUT-2s to match the learning capacity of fewer but larger LUT-$n$s. (2) Fixing $n = 2$ limits FPGA implementation efficiency, since current FPGAs (e.g., Xilinx) use LUT-6 as their logic element.

To overcome these limitations, DWN (Bacellar et al., 2024) was proposed, which employs finite difference and straight through estimator techniques to train multi-layer WNNs. Each LUT determines its gradient backpropagation direction based on the variation in output value caused by flipping address bits. Importantly, DWN allows the interconnections of LUTs to be learnable, meaning that each LUT's $n$ input connections are no longer randomly fixed as in ULEEN. As a result, DWN achieves higher inference accuracy than ULEEN, while also minimizing LUT resource utilization on FPGA implementations. However, the finite difference approach introduces estimation errors, and the simple WNN architecture further limits achievable inference accuracy.

In summary, previous WNNs rely on larger models and higher implementation resources to improve accuracy, posing challenges for deployment on resource constrained edge platforms. To address this limitation, this paper proposes the Convolutional Learnable-Group Weightless Neural Network (CLGN), which introduces the following key innovations:

- **LUT-$n$-Based Convolutional Layers:** We enrich the WNN architecture by introducing LUT-$n$-based convolutional layers, a threshold pooling layer, and multi-channel and multi-threshold input binarization to improve inference accuracy.

- **Learnable GroupSum Connections:** We propose a GroupSum module with learnable and variable connections, enabling more effective logit computation and loss driven optimization.

- **Hierarchical Training Method:** We adopt a two-stage training strategy for CLGN to improve training efficiency. An Multilayer Perceptron (MLP) teacher guides the training of the convolutional and pooling layers, which are then frozen for subsequent WNN and GroupSum training.

We deploy CLGN on FPGA and microprocessor platforms and evaluate it across multiple metrics. Experimental re-

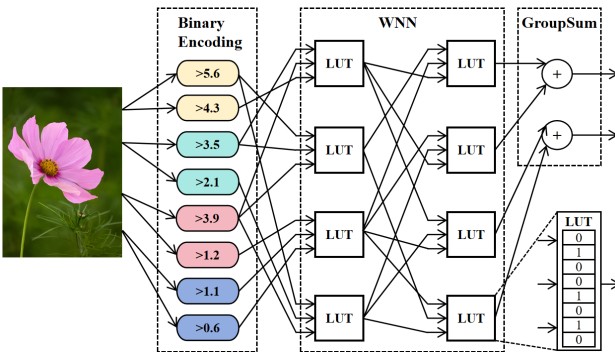

*Figure 1.* Architecture of WNN.

sults show that, by incorporating convolutional operations, CLGN attains higher accuracy with fewer implementation resources and exhibits more favorable accuracy scaling as resources increase, despite the inherent trade-off between inference throughput/latency and accuracy. The related codes are publicly available at https://github.com/astl-xmu/CLGN-paper-related-code.

## 2. Background & Related Work

### 2.1. Weightless Neural Networks

Each neuron in a WNN is implemented using either a single LUT-$n$ or a RAM node. Similar to BNNs, both architectures binarize their weights/activations, thereby eliminating multiplication operations and reducing implementation consumption. However, their connections differ substantially. In BNNs, each neuron remains connected to all outputs of the previous layer, meaning that the implementation consumption remains relatively high. In contrast, as illustrated in Figure 1, the WNN is constructed based on LUT-$n$. For each LUT-$n$, there are exactly $n$ inputs and one output. The output is determined through a lookup process that depends on the specific combination of the $n$ inputs. Consequently, a large number of LUTs are typically required to cover multiple possible $n$ inputs combinations in the dataset, which enhances the model's learning capacity. At the same time, this strengthens its nonlinear representational ability.

Furthermore, since WNNs are implemented using LUTs, they are highly suitable for hardware acceleration based on FPGA, where FPGA's logic elements can directly realize the trained model. Due to their discrete and stochastic nature, LUT-based WNNs perform inference through a table lookup process, which makes effective training challenging. Nevertheless, even under such constraints, existing WNN frameworks have achieved considerable computational efficiency and respectable accuracy (Susskind et al., 2023). Currently, there are two new training solutions for WNNs:

(1) For DiffLogicNet and Conv-DiffLogicNet, the binary

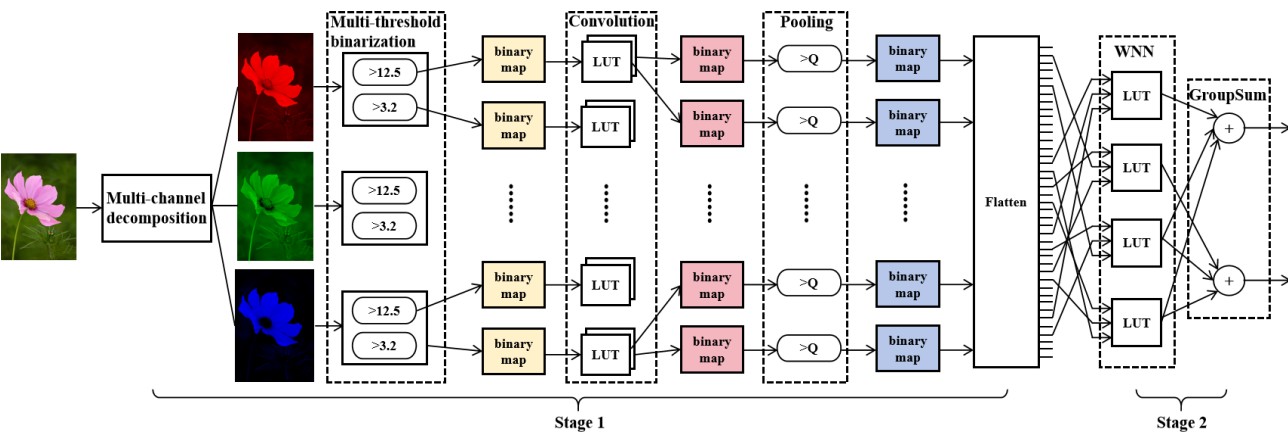

*Figure 2.* Architecture of CLGN.

values are relaxed into probability distributions over all possible Boolean functions and the probabilities are trained. However, $n$ is restricted to 2 due to training difficulty, which necessitates the use of a larger number of LUTs to enhance model capacity. Moreover, when deployed on FPGAs, whose internal logic elements are typically LUT-6, this method consumes a significant number of LUT-6 resources.

(2) For DWN, the finite difference enables each LUT receiving a gradient to update the value at the activated address. It computes the difference between the value at that address and those at neighboring addresses with small Hamming distance, using this difference to determine how the received gradient should be backpropagated. Despite introducing approximation errors, this method supports larger $n$.

## 2.2. Learnable Connections

DWN (Bacellar et al., 2024) introduced the concept of learnable connections for each LUT in WNNs. In traditional WNNs, such as ULEEN, the connections of each LUT are initialized randomly and remain fixed during training. However, since each LUT-$n$ has only $n$ input connections, a large number of LUT-$n$s are required to capture diverse $n$ input combinations. Moreover, these fixed connections cannot guarantee that the selected $n$ input combinations effectively capture the underlying patterns of input variation. To address this limitation, DWN enables each LUT to learn connections by assigning a trainable weight to every possible input connection. During training, these connection weights are iteratively updated, allowing each LUT to identify and retain the most informative $n$ input combinations.

A GroupSum module is appended at the end of the network in WNNs as shown in Figure 1. GroupSum module aggregates the multiple outputs of WNNs to compute the score for each output channel, which is then used to generate the logits. Traditional GroupSum modules, however, lack learnable connections and assign a fixed number of WNN outputs

to each channel. By introducing learnable connections, multiple output channels can share subsets of WNN outputs, enabling collaborative learning among related LUT-$n$s.

## 2.3. Other LUT-Based Neural Networks

In addition to the aforementioned solutions, several other LUT-based neural networks have emerged in recent years. For example, LogicNet (Umuroglu et al., 2020) pre-trains the DNNs and subsequently implements each neuron using LUT-$n$s. However, this approach cannot fully exploit the computational potential of LUT-$n$s, since LUTs are used merely as replacements for DNN neurons. The reason lies in the difference in VC dimension: a single LUT with $n$ inputs has a VC dimension of $2^n$, whereas a neuron with $n$ inputs in a DNN possesses a VC dimension of only $n + 1$. Consequently, LogicNet is unable to construct a WNN architecture with sufficiently high computational capacity.

To further leverage the LUT-$n$s, PolyLUTs (Andronic & Constantinides, 2023) employ feature mappings, and NeuraLUTs (Andronic & Constantinides, 2024) introduce skip connections. During training, these methods increase the internal processing complexity of each LUT-$n$, enabling individual LUT-$n$ to handle multi-bit inputs and outputs. However, since these approaches remain within a LUT-based DNN realization paradigm, the increased LUT complexity primarily serves to emulate DNN style computations. As a result, the benefits in implementation efficiency and learning capacity for WNNs are inherently constrained by the underlying architectural paradigm.

## 3. Methodology

Table 1 summarizes the notations used in this section, and the overall architecture of CLGN is illustrated in Figure 2. CLGN is designed to improve inference accuracy under strict resource constraints through a binary encoding stage

*Table 1.* Notation Definitions.

| Symbol | Meaning |
|--------|---------|
| $n$ | Number of input connections per LUT |
| $m$ | Number of input data channels |
| $S_r$ | Height of convolutional receptive field |
| $S_c$ | Width of convolutional receptive field |
| $S_p$ | Convolutional stride |
| $\Psi$ | Number of binarization thresholds for the dataset |
| $T$ | Number of convolutional kernels for each binary map |
| $P_r$ | Height of pooling receptive field |
| $P_c$ | Width of pooling receptive field |
| $Q$ | Output threshold of pooling layer |
| $C$ | Number of connections per channel in GroupSum |
| $\Phi$ | Number of LUTs in a single layer WNN |
| $N$ | Batch size in training |
| $\eta_1, \eta_2, \eta_{\mathrm{gs}}$ | Learning rates |
| $\tau$ | Temperature parameter controlling GroupSum scaling |

with multi-threshold binarization, LUT-$n$-based convolution and threshold pooling, a single layer WNN, and a learnable GroupSum module. The convolution and pooling layers act as a binary feature extractor, producing structured binary representations. The WNN employs LUT-$n$s with learnable input mappings, and its outputs are aggregated by the GroupSum module to generate class logits via sparse learnable connections. Due to the non-differentiable nature of LUT-based operations, CLGN adopts a two-stage training strategy: the convolution and pooling layers are first trained with a compact MLP teacher and then frozen for efficient training of the WNN and GroupSum modules.

### 3.1. LUT-$n$-Based Convolutional Layers

CLGN introduces the LUT-$n$-based convolution that enriches the WNN architecture with spatial feature extraction while maintaining purely binary representations. Given an input sample, the data are first decomposed according to the number of channels $m$. For image datasets, $m = 1$ for grayscale images and $m = 3$ for RGB images, while for speech datasets, $m$ corresponds to the number of expanded spectral components. Each channel is then binarized using $\Psi$ thresholds, producing $\Psi$ binary maps per channel. The binarization thresholds are determined from the training dataset such that the data distribution is evenly divided into $\Psi$ intervals. For example, given thresholds $\{0.2, 0.4\}$ and a 1D input $[0.3, 0.5]$, the resulting binary maps are $[1, 1]$ and $[0, 1]$, respectively. This multi-threshold binarization preserves relative magnitude information in a binary form and increases representational diversity.

LUT-$n$-based convolution is then applied independently to each binary map. Each convolutional kernel is implemented using a single LUT-$n$, where $n$ input bits are selected from a receptive field of size $S_r \times S_c$. Since a single LUT-$n$ cannot fully cover the receptive field when $S_r S_c > n$, $T$ kernels are assigned to each binary map. These kernels are uniformly and randomly configured to sample different positional combinations within the receptive field, with larger $T$ providing greater spatial diversity. The convolution operation follows a standard sliding window procedure with stride $S_p$, scanning horizontally and vertically across the input. The convolutional responses are not aggregated, resulting in $m \times \Psi \times T$ binary feature maps.

Each binary feature map is subsequently processed by a threshold pooling layer with a receptive field of size $P_r \times P_c$. If the number of activated elements within the pooling window exceeds the threshold $Q$, the pooled output is set to 1; otherwise, it is set to 0. This operation reduces spatial resolution while enhancing robustness to local variations. Finally, the pooled binary feature maps are flattened and fed directly into a single layer WNN. Since LUT-$n$ units in WNNs employ learnable input mappings, no additional data reordering is required. Through this encoding process, CLGN significantly enriches the representational capacity of WNNs while preserving their binary and LUT-based nature.

### 3.2. Learnable GroupSum Connections

CLGN introduces a learnable GroupSum module that enables effective logit computation and loss driven optimization. The single layer WNN consists of $\Phi$ LUT-$n$ units, each producing a binary output based on its learned input combination. These binary outputs are fed into the GroupSum module, which aggregates them into class-level logits. For a classification task with $K$ classes, the GroupSum module maintains $K$ learnable connection matrices, one for each class channel, to learn which WNN outputs are aggregated into each channel (see Appendix A).

To ensure sparsity, only the top-$C$ connections with the largest weights are activated. The selected binary outputs are summed and scaled to form the corresponding class logit. By allowing both the connectivity pattern and the contribution of individual LUT-$n$ units to be learned from data, GroupSum enables more expressive and task adaptive aggregation compared to fixed summation schemes.

### 3.3. Hierarchical Training Method

**Stage 1: Training of convolutional and pooling layers.** The convolutional and pooling layers are first trained under the supervision of a compact MLP. Given an input sample $x$, the output of the pooling layer is denoted as

$$z = f_{\mathrm{pool}}\big(f_{\mathrm{conv}}(x; \Theta_{\mathrm{conv}})\big),$$

where $\Theta_{\mathrm{conv}}$ represents all LUT-$n$ tables within the convolutional kernels of size $S_r \times S_c$ and stride $S_p$. Each LUT-$n$ produces an activation determined by its binary address composed from $n$ input bits. The pooled output $z$ is flattened and passed through the MLP classifier $f_{\mathrm{mlp}}(z; \Theta_{\mathrm{mlp}})$ to obtain logits $\hat{y}$. The training objective of this stage is the standard cross-entropy loss:

$$\mathcal{L}_1 = -\frac{1}{N} \sum_{i=1}^{N} y_i^\top \log\big(\mathrm{softmax}(\hat{y}_i)\big),$$

where $y_i$ is the one-hot label. Gradients from $\mathcal{L}_1$ are propagated from the MLP back to the convolutional LUTs through the pooling layer. Since the pooling operation outputs 1 only when the number of activated elements within its receptive field ($P_r \times P_c$) exceeds the threshold $Q$, only those convolutional outputs contributing to active pooled regions receive non-zero gradients, and the gradient within each activated pooling window is evenly distributed among its active binary elements. Hence, the effective gradient for a given LUT entry indexed by address $a$ is accumulated as

$$G[a] = \sum_{u:\, a(u)=a} \frac{\partial \mathcal{L}_1}{\partial y(u)},$$

where the summation spans all receptive field positions $u$ whose pooling windows are activated. Each LUT entry is then updated as

$$\theta[a] \leftarrow \mathrm{clip}\big(\theta[a] - \eta_1\, G[a],\, [-1, 1]\big),$$

where each entry stores a floating point value and is binarized using zero as the threshold during the forward pass. The clipping range $[-1, 1]$ is applied to maintain stability and logical consistency. The parameters $\Theta_{\mathrm{mlp}}$ are optimized concurrently using Adam. Then, the convolutional and pooling layers are frozen to serve as a binary feature extractor.

**Stage 2: Training of the single layer WNN and Group-Sum module.** After Stage 1, the convolutional and pooling layers produce binary vector $z \in \{0, 1\}^D$ for each input sample. These features are then fed into a single layer WNN composed of $\Phi$ LUT-$n$ units. Each unit $j$ selects $n$ input positions specified by its learnable mapping $M_j$ (see Appendix A) and generates a binary output

$$b_j = \mathbf{1}\big[f_{\mathrm{LUT}\text{-}n}(z_{M_j}) > 0\big].$$

The GroupSum module aggregates these binary responses into class logits. The number of class is $K$, a learnable connection matrix $W_{\mathrm{gs}} \in \mathbb{R}^{K \times \Phi}$ assigns a weight to each connection between the $\Phi$ LUT outputs and the channel of GroupSum. During training and inference, only the top-$C$ connections with the largest weights in each row of $W_{\mathrm{gs}}$ are activated, forming the index set $\mathcal{S}_k (k \in \{1, ...K\})$. The

class logit is then computed as the scaled sum of the selected binary responses:

$$o_k = \frac{1}{\tau} \sum_{j \in \mathcal{S}_k} b_j.$$

The model is optimized using the same cross-entropy loss:

$$\mathcal{L}_2 = -\frac{1}{N} \sum_{i=1}^{N} y_i^\top \log\big(\mathrm{softmax}(o^{(i)})\big).$$

During back propagation, the gradient of $\mathcal{L}_2$ is distributed only to the LUT units that are connected by the current GroupSum mapping and the GroupSum mapping $W_{\mathrm{gs}}$ is updated using the class-wise gradient $\partial \mathcal{L}_2 / \partial o_k$ (see Appendix A). For each LUT-$n$ unit $j$, its gradient is given by

$$l_j = \sum_{k:\, j \in \mathcal{S}_k} \frac{\partial \mathcal{L}_2}{\partial o_k}.$$

Each connected LUT is then updated in two parts: (1) its internal table entries, based on this propagated gradient, and (2) its input mapping $M_j$. Consequently, only the LUTs linked by $W_{\mathrm{gs}}$ receive updates. For each connected LUT-$n$ unit $j$, its table is updated address-wise by hit accumulation and then clipped:

$$G_j[a] = \sum_{i:\, a_j(i)=a} l_j^{(i)},$$

$$\theta_j[a] \leftarrow \mathrm{clip}\big(\theta_j[a] - \eta_2\, G_j[a],\, [-1, 1]\big),$$

where $a_j(i)$ is the address hit by sample $i$ at unit $j$, and $\theta_j[a]$ denotes the $a$-th entry of the $j$-th LUT table. The input mapping of each LUT unit $M_j$ is updated using the unit-wise gradient $l_j$, which aggregates contributions from all classes selecting that unit.

**Objective summary.** The overall optimization alternates between

$$\min_{\Theta_{\mathrm{conv}}, \Theta_{\mathrm{mlp}}} \mathcal{L}_1 \quad \text{and} \quad \min_{\{\theta_j, M_j\}_{j=1}^{\Phi}, W_{\mathrm{gs}}} \mathcal{L}_2.$$

which effectively disentangles binary feature formation from class level relational learning. We provide a detailed analysis of the learnable GroupSum mechanism, including its difference from fixed connectivity, in Appendix F. In addition, the design rationale of the CLGN framework is discussed in Appendix G.

## 4. Experimental Evaluation

We evaluate CLGN on MNIST, KMNIST, SVHN, CIFAR-10, and Speech Commands. These datasets have been widely

used in prior work on BNNs and WNNs, enabling fair and consistent comparisons across architectures. Detailed model and training configurations are provided in Appendix B. We deploy WNNs on two distinct platforms, an FPGA and a microprocessor, to show that CLGN can achieve higher test accuracy while consuming fewer circuit resources or requiring less memory. We select BNNs and existing WNNs with publicly available implementations for comparison. For each WNN baseline, the implementation and accuracy evaluation follow the model size configurations recommended in their original papers.

When deploying BNN and WNNs on FPGA and microprocessor platforms, the trained models must be converted into Verilog and C++ code. To ensure fairness in comparison across different WNN methods, we adopt a unified implementation strategy. Specifically, for the FPGA, all models use the same Verilog coding style, top level interfaces, and data transfer protocols as CLGN, while maintaining consistent module partitioning and forward propagation procedures. For the microprocessor, we employ the same C++ framework as CLGN, including identical buffer organization, data type definitions, and timing methodology.

Regarding input data encoding, each method is configured according to the formats supported in its publicly available implementation (see Appendix C for detailed configurations). BNNs (Courbariaux & Bengio, 2016; Hubara, 2017) directly accept standard floating point inputs. PolyLUT and NeuraLUT support multi-bit inputs, and in our experiments we use the 2-bit representation permitted in their original implementations (thus retaining 2-bit precision per input). ULEEN, DiffLogicNet, and DWN use a unified thermometer encoding (Carneiro et al., 2015) to binarize the inputs. Aside from these necessary format alignments, no additional data preprocessing is applied for any method.

### 4.1. CLGN on FPGA

We deploy the WNN models on a Xilinx Alveo U50 FPGA (AMD), which provides abundant logic resources, including 872K LUTs (LUT-6s), enabling the deployment of relatively large models. In addition, to account for differences in achievable clock frequencies across architectures, each design is operated at the maximum stable clock frequency it can support. To ensure a fair comparison of FPGA resource utilization, all models are constrained to fit within the resource limits of the Alveo U50 FPGA. As shown in Table 2, we report the test accuracy, single inference latency, throughput, power, LUTs usage, and parameter size of the WNN models across five datasets. The configuration details of CLGN are provided in Appendix Table 4.

From the Table 2, it is evident that LGN and CLGN consistently achieve the lowest LUTs usage and smallest parameter size (LGN is derived from CLGN by eliminating the convolutional and pooling layers, while preserving the learnable GroupSum). Although the test accuracy on the more challenging datasets (SVHN and CIFAR-10) still falls short of BNN, CLGN attains the highest accuracy among all WNN approaches other than BNN. These results indicate that CLGN can obtain high accuracy on grayscale images, color images, and audio datasets with convolutional structure, even with substantially smaller models. Compared with CLGN, LGN removes the convolutional and pooling layers and employs thermometer encoding. However, the LUT usage and parameter size remain largely unchanged. In contrast, CLGN achieves a more noticeable accuracy improvement, albeit at the cost of reduced inference speed.

We conduct extended experiments on the CIFAR-10 dataset to analyze different variants of CLGN. CLGN(+) adopts a larger model than CLGN and achieves improved accuracy, outperforming BNNs, at the cost of increased implementation resources; compared with DWN(+) and LGN(+) (which also adopt larger models than DWN and LGN, respectively), it exhibits a larger increase in accuracy relative to the increase in implementation resources.

CLGN($r$) keeps the same model scale as CLGN but does not train the convolutional layers, leading to a moderate accuracy degradation. This indicates that the MLP component effectively guides the LUT-6–based convolutional layers. The limited performance drop can be attributed to the uniform initialization of the LUT-6 parameters, which, together with convolution and threshold pooling, can still produce meaningful binarized maps. CLGN($p$) fully parallelizes the trained CLGN model for implementation, which effectively increases inference throughput and reduces latency; however, the excessive replication of LUT-6 units results in a substantial increase in implementation resources.

However, compared with other WNNs, CLGN requires an additional convolutional computation. Its throughput is noticeably lower than that of other WNN approaches. Despite its reduced FPGA resource usage, it exhibits higher per-inference latency. In the case of BNN, convolutional operations are used only for the SVHN and CIFAR-10 models as specified in the original paper. Due to its very high FPGA resource consumption, BNN must reduce the degree of parallelism in its logic design, which lowers its computational efficiency. Thus, for color datasets, where convolution is also required, BNN exhibits substantially increased inference latency and reduced throughput.

In addition, the BNN models for SVHN and CIFAR-10 contain only a single convolutional layer, which inevitably limits their accuracy. However, this design choice is mainly constrained by the limited on-board FPGA resources required for deployment. Even with only one convolutional layer, BNN still incurs very high FPGA resource consumption and inference latency. Nevertheless, under this setting,

*Table 2.* The trained models (publicly available) are implemented on the Alveo U50 FPGA. LGN is a simplified version of CLGN that removes convolutional and pooling layers and retains the learnable GroupSum; DWN(+)/LGN(+)/CLGN(+) denotes a larger model compared to DWN/LGN/CLGN; CLGN(*r*) denotes a CLGN variant with randomly initialized and fixed convolutional and pooling layers, while CLGN(*p*) denotes a CLGN variant where the convolutional and pooling layers are implemented using fully parallel computation. The corresponding results for Conv-Diff* are taken from the original Conv-DiffLogicNet paper and are included as a reference baseline only.

| Dataset | Model | Test Acc. (%) | Lat. (ns) | Throug. (Samples/s) | Power (W) | LUTs | Param. Size (KiB) |
|---|---|---|---|---|---|---|---|
| MNIST | BNN | 97.71 | 4405.3 | 0.7M | 3.9 | 259070 | 402.4 |
| | PolyLUT | 92.16 | 17.9 | 55.6M | 3.6 | 179238 | 1322.3 |
| | NeuraLUT | 96.51 | 15.8 | 63.1M | 3.6 | 180939 | 1330.3 |
| | DiffLogicNet | 96.79 | 10.3 | 96.8M | 2.4 | 27752 | 14.7 |
| | ULEEN | 95.02 | 12.4 | 80.9M | 2.4 | 23131 | 62.5 |
| | Conv-Diff* | 98.46 | — | — | — | 147000 | — |
| | DWN | 97.93 | 9.5 | 105.2M | 2.3 | 10111 | 64.9 |
| | **LGN** | 98.17 | **8.5** | **117.1M** | **2.2** | **7768** | **35.9** |
| | **CLGN** | **98.76** | 5263.0 | 0.2M | **2.2** | 8975 | 36.1 |
| KMNIST | BNN | 91.54 | 4769.3 | 0.6M | 4.8 | 542102 | 1293.5 |
| | PolyLUT | 78.96 | 17.5 | 57.1M | 3.6 | 179735 | 1322.3 |
| | NeuraLUT | 84.21 | 15.8 | 63.2M | 3.6 | 181627 | 1330.3 |
| | DiffLogicNet | 86.97 | 19.1 | 52.3M | 2.6 | 49038 | 29.3 |
| | ULEEN | 89.22 | 12.8 | 77.9M | 2.4 | 35706 | 93.8 |
| | DWN | 88.86 | 10.9 | 91.4M | 2.4 | 13571 | 99.6 |
| | **LGN** | 90.12 | **8.9** | **112.9M** | **2.3** | **9686** | **58.6** |
| | **CLGN** | **92.63** | 5553.7 | 0.2M | **2.3** | 11510 | 58.8 |
| SVHN | BNN | **81.43** | 49383.6 | 19.9K | 4.2 | 312412 | 802.4 |
| | PolyLUT | 66.02 | 24.4 | 40.9M | 5.5 | 788449 | 5710.5 |
| | NeuraLUT | 75.85 | 19.9 | 50.2M | 4.9 | 706103 | 5110.5 |
| | DiffLogicNet | 66.55 | 24.7 | 40.5M | 4.3 | 363906 | 234.4 |
| | ULEEN | 62.14 | 23.6 | 42.4M | 2.9 | 114807 | 281.3 |
| | DWN | 73.95 | 12.5 | 79.8M | 2.5 | 41932 | 213.0 |
| | **LGN** | 75.97 | **9.1** | **109.6M** | **2.4** | **33532** | **127.4** |
| | **CLGN** | 78.91 | 10204.2 | 126.2K | **2.4** | 38468 | 131.4 |
| CIFAR-10 | BNN | 63.84 | 57613.7 | 17.7K | 5.6 | 798367 | 2068.5 |
| | PolyLUT | 44.93 | 24.4 | 40.9M | 5.5 | 789454 | 5710.5 |
| | NeuraLUT | 49.58 | 19.9 | 50.1M | 5.0 | 705267 | 5110.5 |
| | DiffLogicNet | 55.33 | 24.5 | 40.8M | 4.3 | 364506 | 234.4 |
| | ULEEN | 55.68 | 23.1 | 43.2M | 3.1 | 126926 | 375.0 |
| | Conv-Diff* | 60.38 | — | — | — | 400000 | — |
| | DWN | 56.12 | 16.2 | 61.7M | 2.6 | 48109 | 283.4 |
| | DWN(+) | 57.34 | 20.1 | 49.7M | 2.7 | 77216 | 578.1 |
| | **LGN** | 57.71 | **9.5** | **105.1M** | **2.5** | **38946** | **169.9** |
| | **LGN(+)** | 58.85 | 21.1 | 47.5M | 2.7 | 79375 | 521.5 |
| | **CLGN** | 62.53 | 12657.5 | 101.8K | **2.5** | 41212 | 173.3 |
| | **CLGN(+)** | **65.36** | 16779.2 | 76.8K | 2.7 | 68873 | 276.5 |
| | **CLGN(*r*)** | 59.94 | 12720.8 | 101.3K | **2.5** | 41418 | 173.3 |
| | **CLGN(*p*)** | 62.53 | 19.0 | 52.7M | 4.3 | 373234 | 173.3 |
| SpeechCmd | BNN | 81.49 | 2706.2 | 1.1M | 3.4 | 164326 | 594.8 |
| | PolyLUT | 43.08 | 25.9 | 38.6M | 5.7 | 807541 | 5710.5 |
| | NeuraLUT | 53.24 | 19.9 | 50.1M | 5.0 | 723757 | 5110.5 |
| | DiffLogicNet | 51.19 | 24.6 | 40.6M | 4.6 | 500964 | 234.4 |
| | ULEEN | 72.67 | 22.8 | 43.9M | 3.1 | 135935 | 281.3 |
| | DWN | 78.92 | 10.3 | 96.9M | 2.7 | 57917 | 142.5 |
| | **LGN** | 81.86 | **9.9** | **100.9M** | **2.6** | **50292** | **86.9** |
| | **CLGN** | **82.37** | 95.2 | 10.5M | **2.6** | 50966 | 100.9 |

it still outperforms all basic WNN models.

Overall, CLGN is able to achieve higher accuracy across datasets using significantly fewer LUTs and much smaller models. Moreover, under comparable increases in implementation resources, CLGN exhibits more pronounced accuracy improvements than alternative models. However, this comes at the cost of reduced inference speed due to the inclusion of convolutional operations, which is an inherent trade-off. Consequently, LGN is more suitable when extreme inference throughput or minimal latency is required; however, in edge deployments, model size is often a key factor determining practical deployability.

In addition, since no publicly available implementation of Conv-DiffLogicNet is provided by the authors, we include the results reported in its original paper only as a reference for comparison with LGN and CLGN. Although Conv-DiffLogicNet also supports convolutional operations, CLGN differs substantially from it in several key aspects, including the LUT-6-based convolution kernel, the use of a single convolutional layer, multiple uniform 6-bit samplings, threshold-based pooling, and the training strategy. These design differences enable CLGN to use fewer LUT resources while achieving higher accuracy. Further discussion is provided in Appendix B.2.

### 4.2. CLGN on Microprocessor

We adopt the Raspberry Pi 4B (Raspberry Pi Foundation) as a low cost microprocessor platform (Park et al., 2025). It is equipped with an ARM Cortex-A72 (1.5GHz) CPU and 4GB DDR4 memory. The ARM Cortex-A72 CPU used in the Raspberry Pi is widely deployed in embedded and edge computing devices. Meanwhile, the Raspberry Pi itself is frequently used as an affordable edge inference platform. We directly port the trained models used in the FPGA experiments in Section 4.1 to the Raspberry Pi 4B for evaluation. Since microprocessors do not offer the large scale parallelism available on FPGAs, this setup enables us to further examine the per-inference latency and memory usage on a general purpose processor under identical model structures.

As shown in Table 3, we report the per-inference latency and maximum memory usage of BNN and WNN models across five datasets. Since the models used on the microprocessor are identical to those used in the FPGA experiments, their accuracy, parameter size, and other relevant metrics remain unchanged. The primary differences arise from computational performance and memory overhead: microprocessors lack the massively parallel logic resources of FPGAs, so all computations and data manipulations are executed sequentially by the CPU. Consequently, the per-inference latency increases on the microprocessor.

*Table 3.* The trained models are implemented on the Raspberry Pi 4B platform.

| Dataset | Model | Lat. (μs) | Mem. (KiB) |
|---------|-------|-----------|------------|
| MNIST | BNN | 2077.7 | 427.9 |
| | PolyLUT | **32.1** | 1426.2 |
| | NeuraLUT | 32.5 | 1434.8 |
| | DiffLogicNet | 615.2 | 322.9 |
| | ULEEN | 568.9 | 230.6 |
| | DWN | 289.8 | 151.2 |
| | **LGN** | 144.2 | **112.4** |
| | **CLGN** | 1288.7 | 121.8 |
| KMNIST | BNN | 4436.7 | 1280.5 |
| | PolyLUT | 32.7 | 1426.2 |
| | NeuraLUT | **32.4** | 1434.8 |
| | DiffLogicNet | 1235.4 | 575.6 |
| | ULEEN | 854.8 | 309.8 |
| | DWN | 429.5 | 190.4 |
| | **LGN** | 215.7 | **131.8** |
| | **CLGN** | 1349.4 | 145.3 |
| SVHN | BNN | 2674.3 | 823.5 |
| | PolyLUT | 197.5 | 5918.1 |
| | NeuraLUT | **159.4** | 5304.4 |
| | DiffLogicNet | 3325.3 | 4075.2 |
| | ULEEN | 2555.6 | 785.5 |
| | DWN | 860.7 | 309.5 |
| | **LGN** | 434.9 | **192.2** |
| | **CLGN** | 8711.8 | 241.5 |
| CIFAR-10 | BNN | 7714.9 | 1918.7 |
| | PolyLUT | 207.1 | 5918.1 |
| | NeuraLUT | **156.9** | 5304.2 |
| | DiffLogicNet | 3342.9 | 4077.7 |
| | ULEEN | 3399.2 | 1022.8 |
| | DWN | 1138.2 | 388.1 |
| | **LGN** | 572.2 | **231.8** |
| | **CLGN** | 7392.3 | 284.1 |
| | **CLGN(+)** | 15160.9 | 403.3 |
| SpeechCmd | BNN | 9432.7 | 683.8 |
| | PolyLUT | 203.5 | 5921.8 |
| | NeuraLUT | **159.8** | 5302.8 |
| | DiffLogicNet | 3423.1 | 4079.4 |
| | ULEEN | 2558.8 | 787.1 |
| | DWN | 568.7 | 232.5 |
| | **LGN** | 307.6 | **154.3** |
| | **CLGN** | 1273.8 | 196.3 |

To further reduce memory usage, we apply bit-packed (see Appendix D) representations to the 1-bit weights and activations in BNN/WNN models, which is a common practice. Specifically, multiple binary values are packed into a single 64-bit word, significantly improving memory efficiency.

Although PolyLUT and NeuraLUT incur high memory usage due to their complex LUT constructions, their inference remains fast since the computation is dominated by regular and cache-friendly memory accesses. LGN and CLGN achieve higher inference accuracy with significantly lower memory usage. While CLGN includes convolutional operations and thus exhibits longer per-inference latency than

some WNN models, this latency gap is considerably smaller than that observed on the FPGA. Since all relevant computations and data operations on the microprocessor must be executed sequentially by the CPU, inference latency tends to scale linearly with the number of LUTs. Previous WNN models require more LUTs to achieve competitive accuracy and therefore incur higher inference latency, which narrows the latency difference relative to CLGN.

## 5. Conclusion

This work proposes a new WNN architecture, CLGN, which achieves higher accuracy with fewer LUTs than conventional WNNs on datasets suitable for convolution. Similar to ULEEN and DWN, CLGN supports LUT-6 implementations. Its main innovations include: a LUT-$n$-based convolutional layer with an associated multi-channel and multi-threshold input binarization scheme, and a threshold pooling layer; a GroupSum module with learnable connections; and a two-stage training strategy for CLGN. These contributions collectively lead to significant accuracy improvements. We evaluate CLGN on two grayscale image datasets, two color image datasets, and one audio dataset, comparing it with existing WNNs. All models are deployed on FPGA to assess latency, throughput, power, LUT usage, and parameter size, and on a microprocessor to measure latency and memory usage. Experimental results demonstrate that CLGN achieves higher inference accuracy while requiring fewer LUTs and less memory usage than prior approaches.

## Acknowledgements

This work was supported by the National Key R&D Program of China No. 2023YFB4502703, the National Natural Science Foundation of China under Grant No. U22A2027.

## Impact Statement

This paper presents work whose goal is to advance the field of Machine Learning. There are many potential societal consequences of our work, none of which we feel must be specifically highlighted here.

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

# A. Learnable Connections

## A.1. GroupSum Mapping Learning

GroupSum maintains a learnable score matrix $W_{\text{gs}} \in \mathbb{R}^{K \times \Phi}$. For each class $k$, the active connection set $\mathcal{S}_k$ is determined by top-$C$ selection from the $k$-th row of $W_{\text{gs}}$, and the corresponding logit is

$$o_k^{(i)} = \frac{1}{\tau} \sum_{j \in \mathcal{S}_k} b_j^{(i)}.$$

Let

$$g_k^{(i)} = \frac{\partial \mathcal{L}_2}{\partial o_k^{(i)}}.$$

The GroupSum mapping is updated in a class-wise manner by accumulating the correlation between $g_k^{(i)}$ and the bipolarized LUT outputs $(2b_j^{(i)} - 1)$:

$$\Delta W_{\text{gs}}[k,j] = \sum_{i=1}^{N} g_k^{(i)} (2b_j^{(i)} - 1), \qquad W_{\text{gs}}[k,j] \leftarrow W_{\text{gs}}[k,j] - \eta_{\text{gs}} \Delta W_{\text{gs}}[k,j].$$

After updating $W_{\text{gs}}$, the top-$C$ sets $\mathcal{S}_k$ are recomputed from the updated scores for the next iteration.

## A.2. LUT Input Mapping Learning

Unlike GroupSum which performs top-$C$ selection at the channel level, each LUT-$n$ unit maintains $n$ independent score vectors over the $D$ input dimensions and determines its forward connections by an argmax operation applied separately to each input.

Each LUT-$n$ unit $j$ selects $n$ input positions specified by its learnable mapping $M_j$ and produces a binary output $b_j^{(i)}$. The mapping $M_j$ consists of $n$ score vectors in $\mathbb{R}^D$, one for each input of the LUT-$n$ unit. Given the current GroupSum selection $\{\mathcal{S}_k\}_{k=1}^{K}$, the gradient propagated to each LUT-$n$ unit $j$ is

$$l_j^{(i)} = \sum_{k:\, j \in \mathcal{S}_k} \frac{\partial \mathcal{L}_2}{\partial o_k^{(i)}} = \sum_{k:\, j \in \mathcal{S}_k} g_k^{(i)}.$$

The input mapping $M_j$ of each LUT unit is updated using this unit-wise gradient: only units selected by at least one $\mathcal{S}_k$ receive non-zero updates. Unlike GroupSum, the gradient for a LUT unit may accumulate contributions from multiple classes when the same unit is selected by several $\mathcal{S}_k$.

Concretely, denote the $n$ selected inputs of unit $j$ for sample $i$ as $z_{M_j}^{(i)} \in \{0,1\}^n$. The mapping update is performed by accumulating the correlation between the bipolarized input bits $(2z_{M_j}^{(i)} - 1)$ and the propagated unit-wise gradient:

$$\Delta M_j = \sum_{i=1}^{N} \left(2z_{M_j}^{(i)} - 1\right) l_j^{(i)}, \qquad M_j \leftarrow M_j - \eta_2 \Delta M_j.$$

This update encourages each LUT unit to increase the selection scores of input positions whose bipolar values align with the descent direction induced by $l_j^{(i)}$.

## A.3. Relation and Difference

**Relation.** Both GroupSum and LUT input mappings are optimized using gradients derived from the same loss $\mathcal{L}_2$ and rely on sparse connection structures induced by top-$C$ selection (GroupSum) and $n$-tuple addressing (LUT). In both cases, the mapping update can be written as an accumulated correlation between a local gradient signal and a bipolarized binary vector.

**Difference.** GroupSum mapping learning is class-wise, driven directly by $g_k^{(i)}$ and applied independently to each class channel (row of $W_{\text{gs}}$). In contrast, LUT input mapping learning is unit-wise, driven by the aggregated gradient $l_j^{(i)}$, which may sum contributions from multiple classes selecting the same LUT unit.

# B. Experimental Settings and Scope

## B.1. LGN and CLGN Model Configurations

*Table 4.* LGN and CLGN model configurations for the FPGA and microprocessor experiments. $epoch_1$ denotes the training epochs in Stage 1, and $epoch_2$ denotes the training epochs in Stage 2.

| Dataset | Model | $n$ | $S_r$ | $S_c$ | $S_p$ | $\Psi$ | $T$ | $P_r$ | $P_c$ | $Q$ | $C$ | $\Phi$ | $N$ | $\eta_1, \eta_2, \eta_{gs}$ | $\tau$ | $epoch_1$ | $epoch_2$ |
|---|---|---|---|---|---|---|---|---|---|---|---|---|---|---|---|---|---|
| MNIST | LGN | 6 | - | - | - | - | - | - | - | - | 200 | 2000 | 128 | 5e-3, 5e-3, 5e-3 | 8 | - | 5000 |
|  | CLGN | 6 | 5 | 5 | 1 | 4 | 8 | 3 | 3 | 1 | 200 | 2000 | 128 | 5e-3, 5e-3, 5e-3 | 8 | 500 | 5000 |
| KMNIST | LGN | 6 | - | - | - | - | - | - | - | - | 600 | 3000 | 128 | 5e-3, 5e-3, 5e-3 | 16 | - | 5000 |
|  | CLGN | 6 | 5 | 5 | 1 | 4 | 8 | 3 | 3 | 1 | 600 | 3000 | 128 | 5e-3, 5e-3, 5e-3 | 16 | 500 | 5000 |
| SVHN | LGN | 6 | - | - | - | - | - | - | - | - | 1200 | 6000 | 128 | 5e-3, 5e-3, 5e-3 | 32 | - | 5000 |
|  | CLGN | 6 | 5 | 5 | 1 | 4 | 42 | 5 | 5 | 12 | 1200 | 6000 | 128 | 5e-3, 5e-3, 5e-3 | 32 | 500 | 5000 |
| CIFAR-10 | LGN | 6 | - | - | - | - | - | - | - | - | 1600 | 8000 | 128 | 5e-3, 5e-3, 5e-3 | 50 | - | 5000 |
|  | LGN(+) | 6 | - | - | - | - | - | - | - | - | 4800 | 24000 | 128 | 5e-3, 5e-3, 5e-3 | 150 | - | 5000 |
|  | CLGN | 6 | 5 | 5 | 1 | 4 | 36 | 5 | 5 | 5 | 1600 | 8000 | 128 | 5e-3, 5e-3, 5e-3 | 50 | 500 | 5000 |
|  | CLGN(+) | 6 | 5 | 5 | 1 | 4 | 72 | 5 | 5 | 5 | 2400 | 12000 | 128 | 5e-3, 5e-3, 5e-3 | 60 | 500 | 5000 |
|  | CLGN($r$) | 6 | 5 | 5 | 1 | 4 | 36 | 5 | 5 | 5 | 1600 | 8000 | 128 | 5e-3, 5e-3, 5e-3 | 50 | - | 5000 |
|  | CLGN($p$) | 6 | 5 | 5 | 1 | 4 | 36 | 5 | 5 | 5 | 1600 | 8000 | 128 | 5e-3, 5e-3, 5e-3 | 50 | 500 | 5000 |
| SpeechCmd | LGN | 6 | - | - | - | - | - | - | - | - | 800 | 4000 | 128 | 5e-3, 5e-3, 5e-3 | 24 | - | 5000 |
|  | CLGN | 6 | 1 | 10 | 1 | 4 | 7 | 1 | 10 | 3 | 800 | 4000 | 128 | 5e-3, 5e-3, 5e-3 | 24 | 500 | 5000 |

The configuration details for LGN and CLGN are provided in Table 4. In Stage 1, a lightweight two layer MLP is attached to the pooled binary maps to provide differentiable supervision for the LUT-based convolution and pooling layers. After pooling, the binary output is flattened to a vector $z \in \{0, 1\}^D$, which serves as the MLP input.

For the FPGA and microprocessor experiments, the MLP architecture is:

- a fully connected layer mapping $D \to H$,

- a ReLU activation followed by dropout with probability 0.2,

- a final fully connected layer mapping $H \to K$, where $K$ is the number of classes.

The hidden dimension is set to $H = 12000$ for CLGN and $H = 20000$ for CLGN(+). The MLP parameters are optimized using Adam with the learning rate $\eta_{\mathrm{mlp}}$ ($\eta_{\mathrm{mlp}} = 1e - 4$ for CLGN and $\eta_{\mathrm{mlp}} = 7e - 5$ for CLGN(+)). After Stage 1 is completed, the MLP is discarded; its only role is to provide continuous gradients that guide the LUT-based convolution and pooling layers to learn effective binary feature representations.

## B.2. Convolutional Designs in WNNs and Related Models

In all experiments presented in this work, prior WNN models do not employ convolutional operations. Although Conv-DiffLogicNet (Petersen et al., 2024) reports improvements over DiffLogicNet by using multi-level tree structured logic gates to emulate convolutional kernels and thereby enhance the inference accuracy of WNNs, its implementation has not been publicly released to date, which prevents a fair and effective comparison. Beyond the lack of a public implementation, Conv-DiffLogicNet also does not provide a reproducible mapping from its probabilistic LUT-2 formulation to a concrete discrete LUT implementation. Consequently, any estimation of parameter count or LUT usage would rely on specific realization choices rather than the method itself, and thus would not be meaningful for fair comparison.

Furthermore, even if such approaches were able to improve the learning capability of WNNs, their reliance on LUT-2s requires convolutional behavior to be realized through multi-level compositions of logic gates. This leads to an inevitable increase in overall implementation resources when deployed on FPGA architectures dominated by LUT-6s, making it fundamentally difficult to achieve implementations with low resource usage. In contrast, CLGN constructs convolutional kernels directly using LUT-$n$s ($n = 6$), which allows the trained model to be implemented using the native logic granularity of modern FPGAs, resulting in significantly lower implementation overhead.

In addition, following the original publicly available implementations, BNN (Courbariaux & Bengio, 2016; Hubara, 2017) introduces convolutional layers only on the SVHN and CIFAR-10 datasets, which leads to a noticeable decrease in inference

speed on these two datasets. To ensure fair comparisons, all models are implemented under the same resource constraints of the Alveo U50 FPGA. Consequently, models such as BNN, which require significantly more implementation resources than WNNs, must reduce their model size to fit within the available FPGA resources, resulting in degraded inference accuracy. Even under these constraints, the implementation resources consumed by BNN remain substantially higher.

### B.3. Platform Selection

The experimental platforms are selected to ensure fair and practical evaluation of WNN models. The Alveo U50 FPGA provides sufficient LUT resources to accommodate multiple WNN architectures, enabling comparison based on actual resource utilization instead of estimated values (Bacellar et al., 2024). Raspberry Pi 4B is adopted as a representative edge platform due to its widespread use in practical edge systems for BNN and WNN models (Park et al., 2025). Although its resources exceed those of certain microcontrollers, real-world edge platforms typically execute multiple processes concurrently, and applications with higher accuracy requirements often require increased model capacity. In addition, we measure the memory usage of each WNN model, which more realistically reflects the actual resource consumption of different models.

## C. Data Encoding

BinaryNet is adopted as the BNN baseline in this paper, as it is one of the canonical and widely used BNN architectures. Although the internal weights and activations of the network are binarized, both the model input and the final output remain in floating point format. Consequently, no additional data encoding procedure is required for the BNN, and standard floating point input data can be directly used.

For PolyLUT and NeuraLUT, each input feature is uniformly quantized to 2 bits, resulting in 4 discrete levels per input dimension in this paper. The quantized input values are then directly used as LUT indices, enabling each LUT to map a combination of multi-bit inputs to the corresponding output.

For DiffLogicNet, ULEEN, and DWN, we adopt the thermometer encoding scheme. Each scalar input is encoded using a fixed width of 4 bits, where the bit pattern is determined by the magnitude of the value. Specifically, the encoding follows an ordered activation pattern such as 0000, 1000, 1100, 1110, and 1111, corresponding to progressively larger input values. The bit transition thresholds in the thermometer encoding are fixed by partitioning the training data distribution into four ordered levels, matching the static data driven thresholding principle used in CLGN, and this threshold fitting procedure is consistent with the publicly released DWN implementation.

For CLGN, the $\Psi$ is fixed to 4 during evaluation. Consequently, each real valued feature map is transformed into four binary maps of the same spatial dimensions, based on four predefined thresholds. These settings that 4 discrete levels for PolyLUT and NeuraLUT, 4 bits thermometer encoding for DiffLogicNet, ULEEN, DWN, and LGN, and 4 thresholds for CLGN are chosen to ensure consistency and fairness across all WNN models. Apart from the data encoding settings, all WNN model architectures are implemented following the designs described in their original papers.

## D. Bit-Packed Microprocessor Implementation Details

Due to the support of 64-bit instructions on the Raspberry Pi 4B platform, we represent both the weights and activations in BNNs and WNNs using 64-bit bit-packed formats (`uint64_t`). This representation reduces the overall memory usage of the model while increasing the number of binary elements processed per memory access, thereby improving the overall inference efficiency.

In WNN architectures, in addition to the table values, it is necessary to store the input connectivity indices for each LUT. Since these indices are accessed frequently during forward inference, we avoid applying bit-level compression to the index representation in order to prevent additional decoding overhead. To achieve a balance between storage efficiency and access efficiency, we adopt an adaptive bit-width representation for the connectivity indices. Specifically, `uint8_t` is used when the index range lies within $[0, 255]$, `uint16_t` is used for ranges within $[256, 65535]$, and `uint32_t` is employed for larger input dimensions. This strategy preserves simple and direct index access while effectively reducing the memory cost associated with storing connectivity information.

Table 5 reports the implementation results of CLGNs and prior efficient inference models on the Raspberry Pi 4B under both bit-packed and non-bit-packed settings. In the non-bit-packed implementations, each 1-bit weight or activation is

*Table 5.* Implementation results for CLGNs and prior efficient inference models on the Raspberry Pi 4B under bit-packed and non-bit-packed settings.

| Dataset | Model | Bit. Lat. ($\mu s$) | Bit. Mem. (KiB) | Non. Lat. ($\mu s$) | Non. Mem. (KiB) |
|---------|-------|---------------------|-----------------|---------------------|-----------------|
| MNIST | BNN | 2077.7 | 427.9 | 5988.5 | 2843.9 |
| | PolyLUT | **32.1** | 1426.2 | **32.5** | 5377.5 |
| | NeuraLUT | 32.5 | 1434.8 | 32.8 | 5410.2 |
| | DiffLogicNet | 615.2 | 322.9 | 1103.6 | 436.8 |
| | ULEEN | 568.9 | 230.6 | 531.6 | 683.8 |
| | DWN | 289.8 | 151.2 | 257.2 | 373.6 |
| | **LGN** | 144.2 | **112.4** | 132.7 | **227.5** |
| | **CLGN** | 1288.7 | 121.8 | 1560.9 | 245.2 |
| KMNIST | BNN | 4436.7 | 1280.5 | 11856.8 | 9501.0 |
| | PolyLUT | 32.7 | 1426.2 | **32.9** | 5377.5 |
| | NeuraLUT | **32.4** | 1434.8 | 33.1 | 5410.2 |
| | DiffLogicNet | 1235.4 | 575.6 | 2240.2 | 801.3 |
| | ULEEN | 854.8 | 309.8 | 832.8 | 989.2 |
| | DWN | 429.5 | 190.4 | 395.6 | 526.4 |
| | **LGN** | 215.7 | **131.8** | 199.2 | **303.6** |
| | **CLGN** | 1349.4 | 145.3 | 1622.6 | 321.4 |
| SVHN | BNN | 2674.3 | 823.5 | 18192.8 | 5520.5 |
| | PolyLUT | 197.5 | 5918.1 | 188.7 | 22997.1 |
| | NeuraLUT | **159.4** | 5304.4 | **151.6** | 20581.0 |
| | DiffLogicNet | 3325.3 | 4075.2 | 9056.3 | 5854.5 |
| | ULEEN | 2555.6 | 785.5 | 1793.5 | 2826.2 |
| | DWN | 860.7 | 309.5 | 570.4 | 986.5 |
| | **LGN** | 434.9 | **192.2** | 414.2 | **541.1** |
| | **CLGN** | 8711.8 | 241.5 | 14262.3 | 623.8 |
| CIFAR-10 | BNN | 7714.9 | 1918.7 | 51847.9 | 14318.7 |
| | PolyLUT | 207.1 | 5918.1 | 189.3 | 22997.1 |
| | NeuraLUT | **156.9** | 5304.2 | **150.4** | 20580.5 |
| | DiffLogicNet | 3342.9 | 4077.7 | 9050.5 | 5856.7 |
| | ULEEN | 3399.2 | 1022.8 | 2832.6 | 3740.3 |
| | DWN | 1138.2 | 388.1 | 772.4 | 1287.3 |
| | **LGN** | 572.2 | **231.8** | 548.3 | **693.5** |
| | **CLGN** | 7392.3 | 284.1 | 13473.2 | 764.5 |
| | **CLGN(+)** | 15160.9 | 403.3 | 22335.5 | 1133.8 |
| SpeechCmd | BNN | 9432.7 | 683.8 | 8330.7 | 4574.9 |
| | PolyLUT | 203.5 | 5921.8 | 193.3 | 23010.1 |
| | NeuraLUT | **159.8** | 5302.8 | **155.6** | 20574.8 |
| | DiffLogicNet | 3423.1 | 4079.4 | 9128.4 | 5865.7 |
| | ULEEN | 2558.8 | 787.1 | 1818.1 | 2839.3 |
| | DWN | 568.7 | 232.5 | 520.1 | 698.7 |
| | **LGN** | 307.6 | **154.3** | 288.9 | **401.8** |
| | **CLGN** | 1273.8 | 196.3 | 1516.8 | 508.2 |

represented using `uint8_t`, while each connectivity index in WNNs is stored using `uint32_t`. The results show that the use of bit-packed representations substantially reduces the model memory usage and further leads to a noticeable reduction in inference latency for the BNN, DiffLogicNet, and CLGN models. This is because bit-packing improves data computation and memory access efficiency in BNNs, thereby reducing their inference latency; For DiffLogicNet with small LUT fan-in, bit-packing reduces latency by enabling dense LUT storage with improved cache locality and low per LUT bit access overhead; Meanwhile, it accelerates the GroupSum operations in CLGNs. Since CLGNs involve fewer LUTs, rely on convolutional operations with longer execution time, and are less sensitive to bit-level operations, the overall inference latency is also reduced.

## E. VC Dimension Induced by LUT-$n$

Following (Carneiro et al., 2019), we use the VC framework in which a finite set of input patterns is said to be *shattered* by a learning machine (hypothesis class) if the machine can implement all possible dichotomies (binary labelings) of that set; the VC dimension $d_{\mathrm{VC}}$ is then defined as the cardinality of the largest set that can be *shattered*. Under this definition, (Carneiro et al., 2019) further establishes an exact closed-form VC dimension for a classifier composed of multiple LUT-$n$ units (nodes) when the addressing scheme is fixed and each LUT-$n$ can realize an arbitrary Boolean mapping over its $2^n$ input configurations: if the model contains $\Phi$ such LUT-$n$ nodes, then its VC dimension is

$$d_{\mathrm{VC}} = \Phi\big(2^n - 1\big) + 1.$$

This expression makes explicit that the capacity scales linearly with the number of LUT nodes $\Phi$ and exponentially with the LUT input size $n$; in particular, for the same $\Phi$, the VC dimension of a LUT-6 realization compared to a LUT-2 realization satisfies

$$\frac{d_{\mathrm{VC}}(n = 6) - 1}{d_{\mathrm{VC}}(n = 2) - 1} = \frac{\Phi(2^6 - 1)}{\Phi(2^2 - 1)} = \frac{63}{3} = 21,$$

i.e., increasing the LUT size from $n = 2$ to $n = 6$ increases the dominant VC capacity term by a factor of 21 for fixed $\Phi$ (ignoring the additive constant).

The VC dimension result above characterizes the intrinsic capacity contributed by the LUT-$n$ units under fixed addressing, which form the fundamental nonlinear primitives of CLGN. Beyond this intrinsic component, the overall hypothesis class of CLGN is shaped by additional structured mechanisms, including LUT-based convolution with shared tables, thresholded pooling, learnable input mappings of the WNN layer, and the learnable GroupSum connectivity. These components introduce extra degrees of freedom that adapt the selection and aggregation of discrete features, but do not admit a simple closed-form VC dimension characterization. Importantly, their effect is primarily structural: convolution and pooling impose locality and sharing constraints on feature formation, while learnable mappings and GroupSum reweight and recombine a fixed set of binary responses. As a result, the dominant and explicitly quantifiable contribution to model capacity remains the number of LUT-$n$ units and their input size $n$, whereas the remaining components act to redistribute and regularize this capacity rather than to create unbounded new representational freedom.

## F. Advantage of Learnable GroupSum

**Fixed and learnable connectivity.** In the traditional GroupSum, the sets $\mathcal{S}_k$ are fixed in advance (e.g., sequential or random grouping, $C = \Phi/K$) and remain unchanged during training. In contrast, the proposed learnable GroupSum updates $\mathcal{S}_k$ adaptively through the learnable score matrix $W_{\mathrm{gs}}$. Therefore, the traditional GroupSum can be viewed as a special case of the learnable GroupSum where $\mathcal{S}_k$ is fixed.

**Difference in gradient propagation.** Let $g_k^{(i)} = \partial \mathcal{L}_2 / \partial o_k^{(i)}$ denote the class-wise gradient. The gradient received by LUT-$n$ unit $j$ is

$$l_j^{(i)} = \sum_{k:\, j \in \mathcal{S}_k} g_k^{(i)}.$$

With fixed GroupSum, each $b_j$ belongs to a predetermined $\mathcal{S}_k$, and thus receives gradients from a fixed class channel. With learnable GroupSum, however, the membership of $j$ in $\mathcal{S}_k$ is data driven and evolves during training. As a result, a single LUT output may receive gradients from multiple classes, leading to more flexible and task aligned credit assignment.

**Implication.** The key distinction lies in the structure of $\mathcal{S}_k$. Under the same sparsity constraint $|\mathcal{S}_k| = C$, the learnable GroupSum allows different classes to share LUT outputs through overlapping $\mathcal{S}_k$, whereas fixed GroupSum implicitly enforces a predetermined and typically non overlapping assignment. If a LUT output $b_j$ is selected by multiple $\mathcal{S}_k$, its update is driven by the sum of class-wise gradients. Consequently, such a LUT is optimized as a *shared discriminative feature* across classes, rather than being specialized to a single fixed channel.

This adaptive feature sharing, together with the strictly larger set of admissible $\mathcal{S}_k$ configurations compared to fixed GroupSum, provides a principled explanation for the improved accuracy of the learnable GroupSum, while preserving the simplicity and sparsity of the original design.

Table 6. Accuracy of the LGN model on KMNIST under different values of $C$ ($\Phi = 3000$).

| $C$ | 30 | 150 | 300 (*Fixed*) | 300 | 450 | 600 | 750 | 900 | 1050 | 1200 | 1350 | 1500 |
|---|---|---|---|---|---|---|---|---|---|---|---|---|
| **Accuracy (%)** | 77.98 | 89.88 | 89.03 | 90.09 | 90.33 | 90.12 | 90.04 | 89.38 | 89.11 | 88.03 | 87.79 | 86.31 |

Table 7. Accuracy of the LGN model on KMNIST under different values of $C$ ($\Phi = 6000$).

| $C$ | 60 | 300 | 600 (*Fixed*) | 600 | 900 | 1200 | 1500 | 1800 | 2100 | 2400 | 2700 | 3000 |
|---|---|---|---|---|---|---|---|---|---|---|---|---|
| **Accuracy (%)** | 85.75 | 90.17 | 89.45 | 90.39 | 90.76 | 90.45 | 90.21 | 89.61 | 89.49 | 88.46 | 88.01 | 87.06 |

**Supplementary experiment.** We evaluate the effect of the parameter $C$ on the accuracy of the LGN model on the KMNIST dataset, with the model parameter settings shown in Table 4. The results are summarized in Table 6, where *Fixed* denotes an LGN variant in which the WNN outputs are partitioned into disjoint contiguous groups of size $\Phi/K$, and each GroupSum channel aggregates exactly one such group, corresponding to the conventional GroupSum scheme.

The results indicate that enabling learnable connections in GroupSum improves the accuracy of the LGN model. When $C$ is too small, the LUTs are underutilized, whereas excessively large $C$ leads to increased interference among LUTs. Consequently, only an appropriate choice of $C$ yields optimal performance gains.

In addition, we doubled the parameter $\Phi$ of the LGN model and conducted the same experiments. As shown in Table 7, the larger LGN model exhibits the same trend: achieving higher accuracy requires an appropriate $C$ value, and this optimal $C$ increases as the LGN model size grows. Moreover, although increasing the number of LUT-$n$s expands the combinations of input bits read by the WNN, the resulting accuracy improvement is not as significant as that obtained by introducing a LUT-based convolutional layer.

# G. Design Rationale of the CLGN Framework

## G.1. Design Motivation for Multi-Threshold Binarization

Let $x \in \mathbb{R}^{m \times H \times W}$ denote an input sample with $m$ channels. Each channel is binarized using $\Psi$ thresholds derived from the empirical data distribution. For channel $c$ and threshold index $b$, the binarized output is defined as

$$x^{(c,b)}(u) = \mathbf{1}\big[x^{(c)}(u) > \xi_b^{(c)}\big],$$

where $\xi_b^{(c)}$ partition the value range of $x^{(c)}$ into approximately equiprobable intervals. This operation discretizes continuous inputs into multiple binary representations while preserving relative magnitude information through ordered thresholds.

## G.2. Sampling and Coverage Considerations in LUT-$n$ Convolution

Each convolutional kernel is implemented by a LUT-$n$, whose input address is formed by selecting $n$ binary values from a receptive field of size $S_r \times S_c$. Since $S_r S_c > n$ in general, a single LUT-$n$ cannot cover the entire receptive field. Therefore, for each binary map, $T$ LUT-$n$s are constructed, each corresponding to a distinct $n$-tuple sampled from the receptive field.

The $n$-tuples are sampled uniformly and without replacement from all possible combinations within the receptive field. This design maximizes coverage of the combinational input space while avoiding redundant selections. From a combinatorial perspective, increasing $T$ improves the diversity of sampled input subsets, thereby reducing the bias introduced by any particular positional configuration. As a result, the convolutional layer collectively approximates a richer set of local binary patterns than would be possible with a single fixed sampling scheme.

## G.3. Robustness and Capacity Control via Thresholded Pooling

The pooling layer aggregates binary convolutional outputs over a local region of size $P_r \times P_c$. Given a pooling window $\mathcal{P}$, the output is defined as

$$y = \mathbf{1}\left[\sum_{u \in \mathcal{P}} x(u) > Q\right].$$

This thresholded aggregation suppresses isolated activations and preserves spatially consistent patterns, thereby improving robustness to local noise.

In addition to spatial downsampling, pooling plays a critical role in controlling the dimensionality of the WNN input. By reducing the number of binary features, pooling allows a fixed width WNN to allocate its LUT-$n$s over a broader subset of the remaining feature space. Consequently, the WNN can cover a larger number of distinct $n$-tuple combinations, improving its effective representational capacity under a fixed resource budget.

### G.4. MLP Guided Optimization of Discrete Operators

Direct optimization of LUT-based convolution and pooling layers is challenging due to their discrete and non-differentiable nature. To address this issue, a lightweight MLP classifier is temporarily attached to the pooled binary features during Stage 1. Let $z \in \{0, 1\}^D$ denote the flattened pooled output, and let $\hat{y} = f_{\text{mlp}}(z)$ be the corresponding logits.

Although the convolutional and pooling layers produce binary outputs, the MLP induces a continuous loss surface over $z$ via the cross-entropy objective. Gradients propagated through the MLP and pooling layers provide informative signals that accumulate over LUT entries participating in discriminative patterns. This process enables effective optimization of LUT tables without requiring explicit differentiability of the underlying binary operators.

It is worth noting that uniformly or randomly initialized LUT-based convolutional layers can already achieve reasonable inference accuracy. The MLP guided training further improves performance and stability, though the gain is moderate. This observation indicates that the structural design of the LUT-$n$ convolution and thresholded pooling layers plays a primary role, while the MLP serves as an auxiliary mechanism that refines the learned binary features. Future work will explore more principled training strategies to further enhance the optimization of discrete convolutional operators.

### G.5. Decoupling Discrete Feature Extraction and Classification

After Stage 1, the convolutional and pooling layers define a fixed binary embedding

$$x \rightarrow z(x) \in \{0, 1\}^D,$$

which concentrates task relevant information into a structured discrete representation. Stage 2 then trains a single layer WNN and GroupSum module directly on this embedding. This separation decouples binary feature extraction from discrete decision rule learning, yielding a tractable optimization procedure while preserving the fully discrete nature of the final inference model.

