# OpenReview forum: "Convolutional Learnable-Group Weightless Neural Network"
_ICML.cc/2026/Conference — ICML 2026 regular_

### Official Review · Reviewer_3G7U · 2026-03-07

**Soundness:** 3
**Presentation:** 3
**Significance:** 3
**Originality:** 2
**Overall Recommendation:** 4
**Confidence:** 4

**Summary:**

The paper introduces Convolutional Learnable-Group Weightless Neural Networks (CLGN), a novel WNN design combining LUT-based convolutional layers, a trainable GroupSum aggregation, and a two-phase hierarchical training scheme.  Experiments on multiple datasets, with FPGA and microprocessor implementations, report accuracy, latency, throughput, power, and hardware utilization. Results indicate that CLGN outperforms existing WNN models while requiring fewer LUT resources.

**Compliance With Llm Reviewing Policy:**

Affirmed.

**Final Justification:**

The paper presents CLGN, a LUT-based convolutional WNN with hierarchical training, along with both algorithmic and hardware evaluations. The authors’ rebuttal clarified high latency, hardware trade-offs, and differences from prior convolutional logic models, and provided code access, addressing most of my original concerns.

One remaining suggestion is to include reported metrics from prior convolutional logic papers with an asterisk noting they are copied from the original papers. This would help contextualize CLGN’s performance without requiring reproduction.

Considering the clarifications, I am raising my score.

**Key Questions For Authors:**

1. The reported latency for CLGN is high compared to other WNNs. Can you clarify which architectural or implementation factors cause this overhead?

2. Could you provide more details on hardware choices that influence throughput and latency (e.g., pipelining, memory usage, LUT mapping)?

3. Is there a publicly accessible repository for reproducing your experiments?

4. How does CLGN scale or perform on tasks beyond the presented datasets?

5. How does the model scale to deeper architectures?

**Limitations:**

Limitations are somewhat lacking. Including why the convolution FPGA implementation is slow, or why the models are shallow, and how they scale to deeper architectures would benefit the paper.

**Strengths And Weaknesses:**

## Strengths
- Combining convolutional feature extraction with WNNs is reasonable; evaluation covers both algorithmic and hardware performance.
- Technically sound and explores efficient LUT-based neural architectures.
- Hardware evaluation tables highlight LUT efficiency and edge deployment relevance.
- Well-structured paper with clear explanations of architecture and training procedure.
- Learnable GroupSum connections and hierarchical training offer modest novelty within WNN frameworks.

## Weaknesses
- Experimental comparisons are incomplete; convolutional logic models like Convolutional DiffLogicNet are not included.
- Related work discussion insufficiently positions the method against convolutional logic approaches.
- Latency for CLGN is unusually high without a clear explanation.
- Hardware implementation choices affecting latency and throughput are not fully explained.
- Contribution mainly extends existing WNN architectures; overall methodological novelty is limited.
- Code is not available (although they promise it will), which may hinder reproducibility.

---

> ### Author Rebuttal · Authors · 2026-03-29
>
> We sincerely thank the reviewer for the thoughtful comments and suggestions. For repeated concerns, we also refer to the corresponding replies above.
>
> 1.The comparison is incomplete, due to the absence of convolutional logic models.
>
> To ensure fairness, we restricted our comparisons to relevant models with publicly available code. Although the Conv-DiffLogicNet paper states that the code would be released, its GitHub repository has not been updated, and the implementation details provided in the paper are insufficient for reliable reproduction. This issue is discussed in detail in Appendix B.2: “Beyond the lack of a public implementation, Conv-DiffLogicNet also does not provide a reproducible mapping from its probabilistic LUT-2 formulation to a concrete discrete LUT implementation. Consequently, any estimation of parameter count or LUT usage would rely on specific realization choices rather than the method itself, and thus would not be meaningful for fair comparison. Furthermore, even if such approaches were able to improve the learning capability of WNNs, their reliance on LUT-2s requires convolutional behavior to be realized through multi-level compositions of logic gates. This leads to an inevitable increase in overall implementation resources when deployed on FPGA architectures dominated by LUT-6s, making it fundamentally difficult to achieve implementations with low resource usage.”. In addition, compared with earlier convolutional logic approaches, the convolution mechanism in CLGN differs mainly in the following aspects: (1) The CLGN convolution kernel is constructed using LUT-6s rather than LUT-2s; (2) CLGN uses a single convolutional layer, rather than stacking multiple such layers; (3) CLGN does not use a tree-structured strategy to cover the receptive field, but instead performs multiple uniform 6-bit samplings; (4) CLGN uses threshold-based pooling rather than simple OR pooling; (5) Training. Besides, both CLGN and prior convolutional logic models share a common characteristic with conventional CNNs: they require repeated convolution operations, which inevitably increase latency.
>
> 2.CLGN has relatively high latency, but the explanation is insufficient.
>
> The higher inference latency of CLGN mainly arises from the convolution operations. In the Verilog implementation, we connect submodules through direct logic as much as possible in order to reduce latency. However, the convolution operation must be performed sequentially over the input according to the clock, so the main latency bottleneck lies in the convolution stage. Although latency can be reduced by preparing multiple convolution kernels and performing convolution in parallel, this would significantly increase LUT consumption. This trade-off is also reflected in our ablation study. More broadly, the additional latency introduced by convolution is intrinsically part of the trade-off for achieving higher accuracy. The central goal of CLGN is to obtain competitive accuracy with a relatively small model. We have discussed this point in the manuscript.
>
> 3.The hardware design choices affecting latency/throughput are not sufficiently explained.
>
> In response to Reviewer 2, Comment 2, we clarified how the models are deployed in Verilog. Under a unified implementation style, the differences in latency and throughput mainly arise from the model architectures themselves. Since layers are connected through direct logic whenever possible in order to minimize inference latency, the operating frequency is closely related to LUT usage: in general, the more LUTs a design uses, the lower its achievable clock frequency tends to be, which in turn often leads to higher inference latency. In addition, BNN and CLGN models with convolution layers require extra convolution operations, which also affect latency.
>
> 4.The code link is invalid.
>
> The key training code for CLGN has now been made publicly available at: https://anonymous.4open.science/r/CLGN-paper-related-code-B89D. The remaining code is still being organized.
>
> 5.The discussion of scalability and limitations is insufficient.
>
> Please refer to our response to Reviewer 2, Comment 4.

---

> > ### Author Rebuttal · Reviewer_3G7U · 2026-04-02
> >
> > Thank you for your detailed rebuttal and clarifications. The explanations regarding hardware implementation, latency trade-offs, and convolutional differences are clear, and they resolve most of the concerns I had. Overall, the paper is technically sound and presents the CLGN approach well.
> >
> > One suggestion remains regarding the comparison with prior convolutional logic models, such as Conv-DiffLogicNet. While I understand the code is not publicly available and full reproduction is not feasible, it would still be helpful to include the reported metrics from the original papers (accuracy, LUT usage, etc.) in a table for context. It would be reasonable to add a note or asterisk indicating that these results are taken from the original publications and have not been independently reproduced. This would allow readers to contextualize CLGN’s performance relative to prior work, even if latency, throughput, or power numbers are skipped for those models.
> >
> > In summary, the paper is solid and the clarifications address most of my original concerns. I am therefore raising my score, with the caveat that these prior-model numbers should be included to provide a complete experimental comparison.

---

> > > ### Author Response · Authors · 2026-04-02
> > >
> > > Thank you very much for your constructive suggestion. We are glad that our responses have addressed most of your concerns. We also appreciate your helpful recommendation, which will help us further improve and strengthen the paper.

---

### Official Review · Reviewer_gJa6 · 2026-03-10

**Soundness:** 2
**Presentation:** 2
**Significance:** 3
**Originality:** 3
**Overall Recommendation:** 4
**Confidence:** 3

**Summary:**

The paper proposes a novel weightless neural network architecture. In particular, the paper expands the prior art by proposing a weightless convolutional neural network with learnable groups (CLGN). The validity of the proposed approach is evaluated on small-scale datasets like MNIST, SCHN and CIFAR-10. An FPGA and a Raspberry Pi are considered as deployment target for the proposed method. CLGN is compared on such platforms and such datasets against previous weightless neural network implementations and against Binary Neural Network (BNN).

**Compliance With Llm Reviewing Policy:**

Affirmed.

**Final Justification:**

The replies from the authors adressed most of my concerns

**Key Questions For Authors:**

I have three key questions reflecting the highlighted weaknesses.

1. Can the authors clearly justify why a WNN-based design is compelling with specific quantitative results? Please support this motivation with comparisons against other efficient non-WNN alternatives such as addition-only networks, ternary neural networks, or optimized standard DNN FPGA/MCU implementations. In particular, what is the specific advantage of the proposed approach in terms of accuracy, latency, memory footprint, or energy efficiency?

2. Can the authors justify why the BNN comparison is limited to the 2016 work of Courbariaux and Bengio, and discuss whether this baseline is still representative of the state of the art from the perspectives of training methodology, architectural design, and hardware implementation?

3. Can the authors clarify how the trained models were converted into Verilog and C++ code, including whether this process was manual or automated, whether RTL or behavioral descriptions were used, whether tools such as HLS were involved, and what exactly is meant by “Verilog coding style”? If the process is manual it would be important to highlight the key design choices.

**Limitations:**

There is no dedicated limitations section. I think the limitations could be highlighted better, maybe in a dedicated section. In particular, it would be interesting to discuss what is missing to scale this approach and in general WNN to larger scale applications.

**Strengths And Weaknesses:**

Strengths:

- The paper proposes an interesting extension of the WNN paradigm to convolutions.

- The comparison of the proposed method with prior WNN art clearly show the benefit of the proposed approach.

- The paper is well-written and easy to understand.

Weaknesses:

- The key weakness of the paper is that the need for WNN is not properly motivated. While is clear why having multiplication-less could be beneficial this does not emerge from the results which just compare the proposed CLGN against previous WNN and against BNN. From this reviewer perspective it would be mandatory to also compare with non-WNN designs such as addition only NN, ternary NN and even sota FPGA implementations of standard DNNs. Then, from such comparison it should emerge the clear advantages of WNN. I believe that a task performance (accuracy) drop could be reasonable but I would expect better latency, memory footprint and energy efficiency. If this was already demonstrated in previous WNN-related pubblications this need to be clearly stated and eventually updated with latest BNN/TNN/standard DNN implementations on FPGA/MCU.

- The comparison done with BNN consider the seminal but pretty old work (2016) of Corbariaux and Bengio. Is this the sota for what concern BNN both from a training and architectural standpoint as well as hardware implementation? This point should be discussed.

- The authors stated that "trained models must be converted into Verilog and C++ code" but is not clear how this was done. Writing HDL requires proper architectural thinking, did the author employed a purely behavioral description of the hardware? Did they use tools like HLS? Also is not clear what "Verilog coding style" means.

- The authors should report in Table 2 the max achievable frequency for each design. Also, it would be better to characterize the energy efficiency in terms of Joule per inference.

---

> ### Author Rebuttal · Authors · 2026-03-29
>
> We sincerely thank the reviewer for the thoughtful comments. For repeated concerns, we also refer to the corresponding replies above.
>
> 1.The motivation for WNN/CLGN is not strong enough, and the BNN comparison is outdated.
>
> We have already clarified the detailed BNN configurations in our response to Reviewer 1, Comment 1, and explained our choice of BinaryNet in the response to Reviewer 1, Comment 2. Among low-bit models, WNNs occupy a distinct design point. They rely on LUTs to construct the complete model, and in particular, WNNs based on LUT-6s can be mapped directly onto the computational primitives of FPGAs. As a result, they can be deployed on FPGA with very high implementation efficiency. The objective of WNNs is not to maximize absolute accuracy, but rather to achieve reasonably good accuracy while minimizing FPGA power, basic LUT usage, and hardware consumption on certain other edge platforms. In this sense, WNNs are fundamentally different from other low-bit models. Among low-bit and multiplier-free models, BNNs are currently the most widely studied and most commonly used point of comparison, especially in practical FPGA deployments of WNN-related work. In addition, even some WNN studies not targeting direct FPGA deployment still compare against other low-bit models in simulation and report clear performance advantages; one example is DiffLogicNet.
>
> 2.The Verilog/C++ conversion is unclear.
>
> The final deployment code is implemented manually in advance, with parameterized control over model size, internal table values, and interconnections. For FPGA deployment, we use Verilog directly, in order to avoid the potential resource overhead introduced by HLS. After GPU-side training is completed, the binarized weights/LUT table values and connectivity information are saved. The Verilog modules are written beforehand with configurable structural parameters and are initialized by reading the saved files, which determine the internal stored values and wiring pattern. In other words, the deployment flow does not directly translate Python into Verilog or C++. By “the same Verilog coding style,” we mean that all models are implemented in a as uniform as possible manner to ensure fairness. For example, all implementations use the same external interface and data transfer protocol. Connections between layers are realized as direct logic connections whenever possible rather than through additional pipelining, in order to minimize inference latency. The main exception is between convolutional layers and fully connected layers, where extra registers are required because the convolution operation introduces a mismatch in processing rate.
>
> 3.The FPGA resource report is incomplete.
>
> Please refer to our response to Reviewer 1, Comment 6.
>
> 4.The paper lacks a dedicated discussion of limitations and scalability.
>
> LUT-6-based WNNs are best suited to lightweight tasks. For larger-scale tasks or settings with higher accuracy requirements, models such as BNNs may be more appropriate. The main contribution of this paper is to show that CLGN remains capable of achieving competitive accuracy on lightweight tasks, even when using very small models and requiring only a small number of LUTs for FPGA deployment. We deploy the trained models on actual FPGA hardware and report real measurements, rather than relying only on simulation. Therefore, all tested models are practically implementable. However, even though the Alveo U50 provides substantial resources, deploying larger WNNs remains challenging, and scaling to larger applications would require more FPGA resources. For CLGN in particular, increasing model size can quickly make the resource usage too large for practical deployment. As a result, like DWN, CLGN is more naturally suited to small-scale edge tasks when actual FPGA deployment is required. More sophisticated implementation strategies could partially alleviate this issue, such as trading off throughput and latency for resource reuse, or introducing pipelining across layers and submodules instead of relying on pure logic connections. However, such techniques do not fundamentally remove the scalability constraint. Although non-FPGA deployment platforms may impose less severe resource constraints when scaling CLGN, WNNs based on LUT-6s are specifically designed to align with the computational primitives of FPGAs. Therefore, their advantages are less pronounced on non-FPGA platforms, even though such deployment is still feasible.

---

> > ### Author Rebuttal · Reviewer_gJa6 · 2026-04-02
> >
> > Thanks for the replies to my comments. I raised my score accordingly.

---

> > > ### Author Response · Authors · 2026-04-02
> > >
> > > Thank you very much for your positive feedback. We are glad that our responses have adequately addressed your concerns.

---

### Official Review · Reviewer_1hx5 · 2026-03-13

**Soundness:** 3
**Presentation:** 3
**Significance:** 3
**Originality:** 3
**Overall Recommendation:** 4
**Confidence:** 3

**Summary:**

In this paper, the author argues that prior Weightless Neural Networks (WNNs) and multiplication-free models fail to fundamentally minimize model footprints or achieve high accuracy because they rely on large numbers of parameterized operations or fixed, random connections that capture patterns poorly. To address these issues, the paper introduces the Convolutional Learnable-Group Weightless Neural Network (CLGN), which incorporates LUT-n-based convolutional layers for spatial feature extraction and a GroupSum module with learnable connections to adaptively aggregate outputs into class logits. This architecture is optimized through a two-stage hierarchical training strategy where an MLP teacher guides the non-differentiable layers before they are frozen for WNN training. Experimental results across FPGA and microprocessor platforms show that CLGN achieves superior accuracy and lower memory usage than previous WNNs, though it introduces a trade-off by increasing per-inference latency and reducing throughput.

**Compliance With Llm Reviewing Policy:**

Affirmed.

**Final Justification:**

The author answered all my questions and partially addressed my concerns. I lean towards keeping my original score.

**Key Questions For Authors:**

Please refer to the weakness part.

**Limitations:**

Yes

**Strengths And Weaknesses:**

**Strengths:**
* Deployed on FPGA, and report accuracy, latency, throughput, power consumption, LUTs usage, and parameter size.
* The proposed method directly optimizes the 64 entries of a LUT-6 instead of training a probability distribution over the 2^64 possible Boolean functions. The author effectively reduces the parameter space from 2^64 to just 64 per neuron, enhancing training feasibility for hardware-native widths.
* The author provide more implementation details in the appendix, which is helpful to understand the detailed design of the proposed method.
* Even though the latency performance is not very good on MCU and MCU is not the ideal platform for the LUT-based design. But it is still very good that the author also includes the evaluation on MCU platform.

**Weaknesses**:
* The manuscript states that BNN models were scaled down to fit within the resource constraints of the Alveo U50 FPGA. However, the specific architectural details (e.g., number of layers, channels, and filter sizes) for these reduced BNN baselines are not clearly documented.
* And the author uses the very original BNN baseline. However, there are much more advanced BNN training methods proposed in recent years, which are significantly improving the BNN efficiency and accuracy. The author does not use the SOTA BNN results.
* The author put an anonymous link for the code in the paper. However, the code is not provided there, nor in the supplementary materials.
* While the author provides a comprehensive two-stage training strategy and specifies the number of training epochs (e.g., 500 for Stage 1 and 5000 for Stage 2), the actual training time required to converge on standard hardware is not explicitly reported. Given that WNNs are proposed as resource-efficient alternatives, the computational overhead during the training phase remains unclear, especially with the inclusion of an auxiliary MLP teacher.
* For the reported testing accuracy, are they measured on FPGA or simulated on GPU?
* While the author provides detailed statistics for LUT usage, a comprehensive FPGA resource report should also include the consumption of FFs and BRAMs. It would be great if the author could provide comparisons on those resources.

---

> ### Author Rebuttal · Authors · 2026-03-29
>
> We sincerely thank the reviewer for the valuable comments.
>
> 1.The BNN architecture is not described enough.
>
> In this work, the BNN baseline is based on BinaryNet, with configurations as follows: (1) MNIST: 3 fully connected layers, each with 1000 neurons; (2) KMNIST: 3 fully connected layers, each with 2000 neurons; (3) SVHN: 1 convolutional layer with 20 kernels of size 3×3 per input channel, followed by a 2×2 pooling layer and 2 fully connected layers with 1000 neurons each; (4) CIFAR-10: 1 convolutional layer with 60 kernels of size 3×3 per input channel, followed by a 2×2 pooling layer and 2 fully connected layers with 1000 neurons each; (5) Speech: 3 fully connected layers, each with 600 neurons. For the SVHN and CIFAR-10 datasets, we used only a single convolutional layer in the BNN model for 2 reasons: (1) The resource usage is close to exhausting the available FPGA capacity; (2) Under these settings, the resulting BNN accuracy already exceeds that of all basic WNN baselines.
>
> 2.The BNN baseline is outdated.
>
> The choice of BinaryNet as the BNN baseline was made after careful consideration. BinaryNet is a classical and representative BNN model, and it is also a common reference architecture in FINN. In practice, many FPGA-oriented BNN deployments, especially in work related to WNNs, still rely on the FINN toolchain. For example, the DWN study follows this line. Recent advances in BNNs often improve accuracy by introducing non-binary computations, such as floating-point residuals, or by increasing architectural complexity, such as multi-valued sign functions. While these modifications can indeed improve inference accuracy, they also increase model complexity and make practical FPGA deployment more difficult. In addition, different deployment strategies for such models may introduce substantial implementation variation, making fair comparison more challenging. We aim to obtain competitive inference accuracy with smaller models and lower deployment cost on FPGAs. From the perspective of practical deployment on FPGAs, BinaryNet constitutes a mature and suitable baseline. Besides, to further strengthen the comparison, we additionally evaluated the deeper BinaryNet configurations inspired by commonly used FINN settings. As an example, for SVHN, we considered a BinaryNet with [Conv(20,3×3)+Conv(20,3×3)+2×2 pooling]×3+2 fully connected layers with 512 neurons. It achieved 86.56% accuracy, with 314488 LUTs, 19.4K samples/s throughput, and 112704.3 ns single inference latency. As another example, for CIFAR-10, we considered a BinaryNet with [Conv(60,3×3)+Conv(60,3×3)+2×2 pooling]×3+2 fully connected layers with 512 neurons. It achieved 76.72% accuracy, with 802179 LUTs, 16.9K samples/s throughput, and 133822.5 ns single inference latency. The results show that while BNNs are indeed a strong option when pursuing accuracy, their FPGA deployment cost remains high, and under resource constraints they also significantly increase inference latency.
>
> 3.The code link is invalid.
>
> The key training code for CLGN has now been made publicly available at: https://anonymous.4open.science/r/CLGN-paper-related-code-B89D. The code is still being organized.
>
> 4.The training cost is not reported.
>
> As an example, training CLGN on CIFAR-10 using PyTorch 2.5.1 on an RTX A5000 24GB GPU takes approximately 8 hours for the first stage and 12 hours for the second stage. Our contribution includes a new training method for WNNs, but the focus of the paper is on the deployment characteristics of the trained model, such as inference accuracy and FPGA resource usage. The proposed two-stage training procedure is introduced to make the non-differentiable modules trainable, and its cost is therefore a one-time offline training cost. The training hyperparameters are reported primarily to facilitate reproducibility after code release, rather than to imply any fixed hardware requirement.
>
> 5.The platform used to measure test accuracy is unclear.
>
> The reported inference accuracy is measured on GPU and the hardware-related metrics are obtained from models implemented on FPGA.
>
> 6.The FPGA resource report is incomplete.
>
> We have reported all the key metrics for WNN deployment on FPGA. The deployed models use very little of certain FPGA resources such as FFs, and the WNN models in particular do not use DSP or DRAM. In our setting, LUT usage is the dominant hardware cost for WNN deployment on FPGA, and therefore LUT consumption was the primary resource metric reported, as in other prior work based on actual FPGA deployments. Regarding the FPGA maximum frequency and energy: although the maximum frequency was not explicitly listed, all models were deployed under a fair setting in which each design was configured to run at its maximum stable clock frequency, rather than at a fixed low frequency, in order to obtain the best achievable throughput and single latency. In addition, we reported the power values directly obtained from Vivado.

---

> > ### Author Rebuttal · Reviewer_1hx5 · 2026-04-03
> >
> > The author answered all my questions and partially addressed my concerns. The accuracy is GPU-based, and the BNN structure is not SOTA (it is true that FPGA implementation can be complicated).

---

> > > ### Author Response · Authors · 2026-04-03
> > >
> > > Thank you for carefully considering our rebuttal and for your thoughtful feedback. We are glad that our response helped clarify the paper. We hope the rebuttal also made the practical motivation for studying this design under FPGA-oriented constraints clearer. Thank you again for your time and consideration.

---

### Decision · Program_Chairs · 2026-04-30

**Decision:**

Accept (regular)

**Comment:**

After considering the reviews, rebuttal, and overall discussion, I believe the paper meets the bar for acceptance. While some concerns were raised during review, they do not substantially alter the overall positive assessment of the submission.